# A Perspective: Challenges in Dementia Research

**DOI:** 10.3390/medicina58101368

**Published:** 2022-09-28

**Authors:** Mark Stecker

**Affiliations:** Fresno Institute of Neuroscience, Fresno, CA 93720, USA; mark@finsneuro.org; Tel.: +1-516-478-8304

**Keywords:** dementia, Alzheimer’s disease, big data, neurophysiology, imaging, biomarkers

## Abstract

Although dementia is a common and devastating disease that has been studied intensely for more than 100 years, no effective disease modifying treatment has been found. At this impasse, new approaches are important. The purpose of this paper is to provide, in the context of current research, one clinician’s perspective regarding important challenges in the field in the form of specific challenges. These challenges not only illustrate the scope of the problems inherent in finding treatments for dementia, but can also be specific targets to foster discussion, criticism and new research. One common theme is the need to transform research activities from small projects in individual laboratories/clinics to larger multinational projects, in which each clinician and researcher works as an integral part. This transformation will require collaboration between researchers, large corporations, regulatory/governmental authorities and the general population, as well as significant financial investments. However, the costs of transforming the approach are small in comparison with the cost of dementia.

## 1. Introduction

A total of 115 years has passed since Alois Alzheimer [1] published his paper on dementia, in which he made the optimistic statement: “A histological examination will enable us to determine the characteristics of some of these cases. This process will gradually lead to a clinical distinction of specific illnesses”. However, despite subsequent massive clinical and basic science research [2,3], it remains difficult to identify and diagnose dementia in the early stages, or to develop a disease modifying treatment.

Now is the time for novel approaches. Historically, new and exciting ideas have arisen out of attempts to answer seemingly simple challenges. For example, the X-prizes [4], Hilbert’s 10 problems in mathematics [5], the structure of DNA [6] or the electromagnetic spectrum emitted by a heated object [7], have all generated completely novel and unexpectedly important developments.

The purpose of this paper is to propose, in the context of current knowledge, specific challenges regarding dementia that may stimulate controversy and specific research. Many elements of these challenges have already been addressed to some degree, but the full promise of each challenge cannot be met without integrating multiple techniques and ideas.

## 2. Specific Challenges

### 2.1. Challenge 1: Optimizing and Quantifying the Patient Evaluation

There are many types of dementia, which can have different etiologies, symptoms and prognosis [8,9]. The first step in analyzing dementia must be a widely available, comprehensive, yet concise assessment of each individual, which can be applied serially [10] in both cognitively healthy persons and patients with clear dementia. It must include multiple components including those listed below.

#### 2.1.1. Quantification of Behavior

##### Improved Analysis of Standard Behavioral Testing

The evaluation for cognitive impairment begins at the behavioral level [11]. The problem is that cognitive tests are naturally dependent on many factors, such as age [12,13], race, gender, education, language [14], IQ and experience [15], as well as medical factors such as sleep [16,17], pain and many others. There are two solutions to this problem, one is to pick a test or set of tests and then study how its results change with all of these factors [18]. Although this a very reasonable approach, it is complex because the number of factors that influence test results, even in normal people, will be large and difficult to predict. This results in large test variances for individuals and reduces the precision of the assessment. A second solution is to start with a number of tests including quantitative psychophysical tests (that may be less dependent on education and language than traditional paper and pencil tests) [19,20] and use statistical techniques to find a limited combination of these test elements, which is minimally dependent on the factors that influence cognition in cognitively healthy people. This is not enough, however, because the ideal test should not only yield results that were similar in cognitively healthy people, but also be sensitive to the different phenotypes seen in cognitive disorders of differing etiology.

##### Data Mining/Extraction Techniques

Important techniques that might provide additional information not available in the standardized paper and pencil and psychophysical testing, are modern data mining techniques to extract features in a video [21] or audio recordings of individuals [22]. The use of data available in social media [23] could also provide important information.

##### Other Testing

Other important non-neuropsychological clinical tests that can provide insight into dementia involve eye movements [24,25], retinal function [26], gait [27], olfaction [28,29], taste [30] and hand movements [31].

#### 2.1.2. General Medical Conditions

No assessment of dementia is complete without a knowledge of the patient’s general medical conditions. This would have to include all elements of the traditional history and physical, as well as laboratory testing for common systemic conditions that could affect the brain such as vitamin B12, thyroid function, etc. The National Alzheimer’s Coordinating Center’s Uniform Data Set [32] contains additional data that is also important.

#### 2.1.3. The Neurological Exam

A full description of the patient would also include the results of the general neurologic examination as it provides information on strength, sensation, gait, reflexes, and cranial nerves that are important. Many quantitative methods have been proposed to calculate this [33,34] and the selection of the optimal data variables is critical. At the present time, specific examinations are used for each disease. This approach is useful once the pathophysiology of a disease is clear but for dementia, a wider net must be cast.

#### 2.1.4. Imaging

The results of brain imaging are critical to any full patient description. Clearly, MRI and/or CT images of the brain should be obtained if possible [35,36]. However, a comprehensive evaluation should include the possibility of obtaining perfusion imaging [37] as well as CT angiogram and MR angiogram images [38] and metabolic PET scans [39]. Amyloid and tau imaging [40] are also important, as is functional MRI [41,42]. In particular, functional imaging coupled with electrophysiology can provide information about the brain connectome [43]. The Alzheimer’s Disease Neuroimaging Initiative (https://adni.loni.usc.edu/, accessed on 24 September 2022) has made great progress in this direction, but this pioneering work is just a beginning, and must be extended.

#### 2.1.5. Electrophysiology

Much information on the function of brain networks is available in neurophysiologic studies, such as EEG and evoked potentials in dementia [44,45,46,47] and should be included in the patient database when available

#### 2.1.6. Serum/CSF Biomarkers

Various biomarkers [48] provide valuable information on diagnosing and understanding the mechanisms underlying dementia [49,50]. Some of these are neurofilaments [51], Aβ42 [52,53], tau [54,55], GAP-43 [56], neurogranin [57], trem2 [58], neuron-specific enolase [59], YKL-40 [60], and neuroregulin [61] among others. These also include exosomes and microRNA, which contain important information about the state and function of cells in the central nervous system [62,63,64].

#### 2.1.7. Neuropathology/Histology/Electronmicroscopy

When available, understanding the microstructure of the brain and the individual neurons can provide vital information about the mechanisms of cognitive decline in dementia [65,66,67]. This includes standard histology and immunopathology [68,69]. However, electron microscopy provides crucial information regarding the state of subcellular organelles and structures [70,71].

#### 2.1.8. Omics

Geneomic, proteomics, metabolomics and other similar approaches have been proposed as valuable markers in the study of dementia [72].

##### Genomics

Knowledge of the individual genome [73,74], as well as epigenetic markers of expression [75], are also critical in understanding dementia.

##### Proteomics

In-depth proteomic profiling to identify diagnostic and prognostic markers and gain understanding of complex pathophysiologic mechanisms [76].

##### Denomics

This is the study of the demographic factors that may influence health outcomes including diet [77], education, income, age, exercise and other activities. It also includes exposures to various environmental factors included in the exposome [78].

##### Metabolomics

These studies [79,80] can find patterns of metabolites in the blood that could correlate with dementia, and may thus provide insight into mechanisms.

#### 2.1.9. Managing the Data

What is proposed above is an enormous amount of information, but starting with a database that is too limited is problematic:As more is learned about the different dementias, it may be that factors initially thought of low importance could become increasingly significant.More data will help find the most effective tools for diagnosing and distinguishing among the different dementias.More detailed knowledge will help build model systems that better reflect each type of dementia.Increases the chance of unique new discoveries.Data must be collected longitudinally over time.

#### 2.1.10. Making It Practical

It is easy to conceive of such a large database, but the key to this challenge is to bring together the elements to make it practical:Funding.Government and regulatory agency buy in.“Assuring beneficence, justice and respect for all persons involved” [81].Allowing individuals control over the use of private information.Data storage, access and availability. It is important that elements of the database be available at multiple levels, allowing maximal researcher access at various levels without compromising private information. Allowing general access to elements of the database can allow crowd-sourcing [82] that can lead to new insights.Data analysis. New statistical methods and computing methods will need to be developed to analyze the data, including machine learning [83,84,85], as well as other techniques [86,87].Large patient number. Using a large data set with many dependent variables requires a sizeable number of patients in order to find patterns in the data.

### 2.2. Challenge 2: Quantifying Normal Aging

There are many biologic processes that result in a dementia phenotype. Some of these processes arise from causes outside of the neuronal/glial networks responsible for cognition, such as stroke, infections, trauma and metabolic disorders. Some processes causing progressive cognitive impairment are pathologic, but some may be inherent to the normal brain. Although it is easy to recognize pathologic brain function when it is severe, in order to identify the initial steps in the progression of dementia, it is critical to have very precise definitions of normal function at each age. A database such as the one described under Challenge 1 provides the basis for comparing changes during normal ageing and dementiaa, and allows us to ask more specific questions.

#### 2.2.1. Changes over Time

In the absence of any pathology, what is the fate of the neuronal/glial networks over time? This needs to be determined from many viewpoints including: behavioral [88,89,90,91], imaging [92,93,94,95], exosomes [96], metabolomics [97] and molecular biologic [98].

What is the time course of this change? There may in fact be multiple time dependent changes in different variables. Which variables demonstrate the first changes?What mechanisms underlie the changes in normal elderly patients?Are any of these “normal aging” changes seen in humans, also seen in animals and isolated neurons? Are the time courses of the changes the same in each system?Are there natural processes, exposures or genetic factors in humans or animals that ameliorate or accelerate these time dependent changes?Beware overzealous extrapolation from animals to humans [3].

#### 2.2.2. Model Systems

Eventually, progress toward understanding normal brain ageing and dementia will be facilitated by robust model systems. Model systems that explain only a very few of the changes will not be as valuable as systems that explain many different changes. Models can be based in animals (with caution) [99], in vitro systems [100], mathematical or computational [101,102]. These models will be critical to exploring and creating new ideas.

### 2.3. Challenge 3: Quantifying Pathologic Aging

The same techniques used to study and quantify normal aging can be applied to patients who have diagnoses of different types of dementia:Use the existing data to refine the definitions of the various dementia types [103,104,105,106,107,108].Follow-up longitudinal data over time to find the first difference noted between normal brain aging and pathological aging in patients eventually diagnosed with dementia [109,110,111,112]. This will form a more effective focus for disease modifying treatment than changes that occur late in the illness.What are the characteristics of the nervous system that are different (see Challenge 2) at the onset [113] of pathological aging?What is the detailed temporal relationship between amyloid and tau pathology, and the various biomarkers and behavioral changes in the database [114,115,116]?

### 2.4. Challenge 4: Building New Model Systems for Dementia

There has been much work on animal models [117,118,119,120], cellular models, [121,122,123,124,125] and computational/mathematical models [126,127,128,129] of dementia. However, none of these has captured all of the critical elements involved in the pathogenesis of the dementias and so cannot generate the sought-after answers. With the additional information obtained as part of these challenges, more effective models capturing more of the critical elements of dementia can be created, that will better serve the goal of understanding and treating dementia.

The complexity of the human brain is such that it may not be possible to immediately find a single model. Thus, it is necessary to have multiple overlapping models beginning at the smallest scale. There has been much work on the dynamics of protein folding and molecular dynamics [130,131,132] that has been stretched to the organelle level [133]. Beginning with this, models of the relevant aspects of single cell behavior [134,135] can be created and matched to cellular models. The interplay between predictions and observations at this level can help optimize modelling at this low level. Subsequently, modelling using organoids and tissue slices can be used and compared to mouse rat and primate models, all of which will be compared to humans in various ways.

This requires large scale collaborations between scientists with different backgrounds using different techniques.

### 2.5. Challenge 5: Search for Factors That Modify the Trajectory of Dementia Related Changes

The data generated by the previous challenges forms the substrate to generate hypotheses and test potential treatments for dementia. The model systems can be used to help choose the most appropriate molecules and doses, while predicting side effects of treatment before applying them to humans. Once potential therapies reach the level of human trials, the comprehensive data sets will be the key to understanding outcomes and refining future therapies. In addition, analysis of the demographic and exposure data in the context of multivariable outcomes will be key to using the natural experiments resulting from different genetic and environmental variables to look for potential therapies that can be tested. This requires large numbers of patients and modern computational techniques. Maximal data access that does not compromise individual privacy must be allowed to maximize finding important trends in such a large database.

## 3. Discussion

The costs of dementia at the personal and family level are incalculable, but the global financial costs are estimated to be on the order of USD 3 trillion [136] yearly. By comparison, the total yearly NIH budget at USD 45 billion dollars is only 1.5% of this number, and the operating expenses of a huge corporation such as Amazon are on the order of USD 100 billion each year, or 3% of the costs of dementia. Although the cost of solving the challenges proposed in this paper (Table 1) is huge, they can likely be met with resources of that magnitude, if they are well organized. Despite the cost, the reward is so large that such an investment is warranted.

## 4. Conclusions

Finding a cure for dementia has so far proven intractable using current scientific models. It is now time to pursue a new model driven by large scale collaborations, not only between researchers and clinicians, but also including large corporations and world governments as partners. In addition, involving the general population in decisions about data use and crowd-sourcing analyses on big data will be critical elements of this new approach.

## Figures and Tables

**Table 1 medicina-58-01368-t001:** Summary of Challenges in Dementia.

Challenge	Details	Cost	Difficulty	Comment
Optimizing and Quantifying the Evaluation	Develop and maintain huge clinical databases with clinical, imaging, molecular and biochemical data for large numbers of patients.	++++	++++	Although difficult, this is the necessary pathway to progress in diagnosing, understanding the mechanisms of and treating dementia.
Quantifying Normal Aging	Without a comprehensive understanding of normal brain, aging it is impossible to identify the events that initiate the downward decline we see in dementia.	+++	+++	Once the first challenge has been met, this is much simpler but still requires substantial resources due to the need for long term studies.
Quantifying Pathologic Aging		+++	+++	
Building New Model Systems		++	++	The key is collaboration between teams of scientists with different backgrounds ranging from quantum and statistical physics, chemistry, biochemistry and systems biology to clinical care.
Search for factors that modify the trajectory of dementia related changes		++	++	Once the database is established appropriate computational resources need to be available to allow for analysis.

## Data Availability

Not applicable.

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
