# Peer review of "A Perspective: Challenges in Dementia Research"

_medicina, 2022, doi:10.3390/medicina58101368_

Round 1

Reviewer 1 Report

The Manuscript titled " Challenges in Dementia” by Mark Stecker provides a summary of challenges in the field of dementia. I suggest expanding the manuscript would be desirable and will make it more useful because in many sections, the provided information is too brief and mostly looks like a list of elements.

Author Response

                                                                                                                                                    9/18/22

Editor, Medicina

For: Special issue: Commemorative Issue Celebrating the 20th Anniversary of the Alzheimer’s Foundation of America: Understanding and Treating Alzheimer’s Disease

Dear Editor:

            My responses to the reviewers are in red below.  The comments of the reviewers are in black.

            I thank the reviewers for their comments on the paper originally titled “Challenges in Dementia” by Mark Stecker. 

            I agree with the reviewers that the fundamental issue is that this paper is NOT a research article, a literature review or a case report but one individual’s perspective on research that needs to be done in the future in order to make substantial progress in finding cures for dementia.  I will leave it up to the editors to determine the article type for the purposes of publication. In my view, it is more of an editorial/opinion piece that is very appropriate in a special issue commemorating the Alzheimer’s Foundation of America which is focused on ameliorating and curing dementia.  I like Reviewer 2’s suggestion that the title be changed to reflect this.  Thus the title was changed to “ A Perspective: Challenges in Future Dementia Research”. 

            The reviewers both comment on the outline/telegraphic style of the paper.   This is done on purpose to provide an outline of significant steps that (In my opinion) need to be taken in order to attack the problem of dementia that is extremely complex and has resisted much scientific research to date.   It provides specific “challenges” that can be tested and explored in the future.  Because of the specific nature of the bullet points (challenges), researchers can then: “ comment on”, “agree with”, “disagree with”, or provide new research on each of the topics.  The hope is that this can be a focus for further discussion not a rigorous statement of what is currently known. 

            In addition to changes made specifically on the suggestions of the reviewers, additional minor changes were made throughout the manuscript to enhance clarity.

Reviewer 1

The checklist included the annotations:

1-“Does the introduction provide sufficient background and include all relevant references?”—must be improved;

            I have enlarged the introduction with more references to make the motivation and type of paper clearer.

Specific comments:

The Manuscript titled " Challenges in Dementia” by Mark Stecker provides a summary of challenges in the field of dementia. I suggest expanding the manuscript would be desirable and will make it more useful because in many sections, the provided information is too brief and mostly looks like a list of elements.

            As this paper is NOT a literature review and is intended to be a list of specific challenges in dementia research placed in the context of current clinical and research, I continue to feel that a format with specific bullet points rather than a longer traditional manuscript is more appropriate.  If this were a traditional review paper, I would agree with the reviewer that much more detail would be needed but I think more detail would obscure the needs to look into specific issues that remain unknown

Reviewer 2

Specific Comments:

  1. It is advised to specify the title more. Also, the type of the study should be mentioned in the title. E.g., Narrative Review; Challenges in the quantification of dementia, …

I think that this is a critical observation as the paper is not a traditional literature review or research article.  I have modified the title as suggested to clarify that this.

  1. The type of the manuscript is not an “Article” is a type of literature review.

I will defer to the editors to help classify this paper as it does not fit perfectly as an “Article”, “Literature Review”, or “Case Report”.

  1. The author should include more specific information in the abstract. Remember that in most indexations, only the abstract is provided.

As suggested by the reviewer the abstract has been modified to include more information.

  1. Please, include five keywords.

As suggested, I have added keywords.  There are now 6 keywords

  1. Introduction should have a background for the manuscript. Moreover, the introduction should explain “why this manuscript’s idea is important.” At least three complete paragraphs for better comprehension.

As suggested, the introduction is now 3 paragraphs long. I think these describe the motivation behind the paper and its outline much better than the original introduction.

  1. The grammatical English should be revised. Also, there are some mistyping. E.g., “Challenge 1-Optimizing”

I have spell and grammar checked the document.  The specific instance noted by the reviewer was appending a name to the first challenge.  I agree that this would be clearer if instead of a hyphen a colon were placed between the identification of the challenge and its descriptive name.  This change has been made

The present manuscript appears to be telegraphic. It is advised to reduce the number of subchapters and provide a description. The manuscript should be thoroughly revised regarding style to provide more discourse language. 

As I have commented above, this is not a literature review but a series of discrete challenges and so I think it is more appropriate to keep the current style.  I hope that changing the title and the introduction motivate this more clearly. 

  1. Discussion should be separated from the conclusion.

Done as suggested.

After the challenges’ localization, a small discussion is advised for every challenge. Moreover, the inclusion of future perspectives would significantly impact the quality of the manuscript.

I feel that the paper is oriented toward future perspectives.  That may have been obscured by inadequacies of the title abstract and introduction which have been substantially modified as suggested.

Reviewer 3

An intersting topic on dementia however the manuscript has serious methodological flaws (study design, results presentation, poor discussion).

As noted in the response to reviewers 1 and 2 there is no study design as this is not a research paper and there are no results since this is not a literature review.  I think that the changes made explain more that this is a perspective for future research. 

Introduction: need to be report more scientific background with appropriate references

Changes made.

2.1.1.3-Other Testing: there several other tests, please add them.

            I would be glad to add other clinically based tests felt important by the reviewer if they were specified.   I have taken the liberty of adding two more here. 

2.1.5-Electrophysiology: which are the findings in dementia

There are a large number of parameters that have been analyzed over time including EEG spectral analysis, entropy analysis, event related potentials, signal complexity measures, connectivity measures, various evoked potentials.  Reviewing the results of each type of analysis would be a large paper in itself and so the reader who needs more information can find it in the references.  Again, this is not a review paper but a perspective on the  challenges that need to be overcome in order to make progress in dementia research. 

2.1.6-Serum/CSF Biomarkers:  there are more biomarkers ...

As above, there have been a huge number of potential biomarkers studied in the CSF and the serum and the reviewer has not suggested specific biomarkers felt to be most important.  Again this is not a comprehensive review but I have added 6 others athough there remain many more.

2.1.8 Omics [56]:? please explain,

There are many “omics” studies and this is a caption head.  In order to improve clarity a short phrase was added here.   

2.1.8.4-Metabolomics [63] [64]: please explain more,

As suggested, additional clarification regarding the potential value  of metabolomics studies are noted.

Discussion: please summarized the main challenges in dementia? limitation/ advantages??? future recomentations for research in dementia?

Recommendations for research are noted throughout each challenge but the conclusion and discussion have been modified to indicate that the most important issue is large scale collaborations with more funding.

Sincerely,

Mark Stecker, MD, PhD, FAAN, FACNS, FASNM

Reviewer 2 Report

1. It is advised to specify the title more. Also, the type of the study should be mentioned in the title. E.g., Narrative Review; Challenges in the quantification of dementia, …

2. The type of the manuscript is not an “Article” is a type of literature review.

3. Abstract. The author should include more specific information in the abstract. Remember that in most indexations, only the abstract is provided.

4. Please, include five keywords.

5. Introduction should have a background for the manuscript. Moreover, the introduction should explain “why this manuscript’s idea is important.” At least three complete paragraphs for better comprehension.

6. The grammatical English should be revised. Also, there are some mistyping. E.g., “Challenge 1-Optimizing”

The present manuscript appears to be telegraphic. It is advised to reduce the number of subchapters and provide a description. The manuscript should be thoroughly revised regarding style to provide more discourse language. 

7. Discussion should be separated from the conclusion.

After the challenges’ localization, a small discussion is advised for every challenge. Moreover, the inclusion of future perspectives would significantly impact the quality of the manuscript.

Author Response

(The authors gave the same response as above.)

Reviewer 3 Report

An intersting topic on dementia however the manuscript has serious methodological flaws (study design, results presentation, poor discussion).Introduction: need to be report more scientific background with apropriate references, 2.1.1.3-Other Testing: there several other tests, please add them.2.1.5-Electrophysiology: which are the findings in dementia,2.1.6-Serum/CSF Biomarkers:  there are more biomarkers ...2.1.8 Omics [56]:? please explain, 2.1.8.4-Metabolomics [63] [64]: please explain more,Discussion: please summarized the main challenges in dementia? limitation/ advantages??? future recomentations for research in dementia?

Author Response

(The authors gave the same response as above.)

Round 2

Reviewer 2 Report

The topic discussed in the manuscript is undoubtedly significant to the scientific literature regarding dementia. However, the present manuscript can easily lead to misunderstandings and possible future complications. In the reviewer's opinion, the manuscript should be modified accordingly to describe the topics resembling a literature review. The authors should avoid telegraphic language.

Author Response

                                                                                                                                                    9/20/22

Editor, Medicina

For: Special issue: Commemorative Issue Celebrating the 20th Anniversary of the Alzheimer’s Foundation of America: Understanding and Treating Alzheimer’s Disease

Please find enclosed an article entitled A Perspective: Challenges in Dementia Research.  This is submitted for the special issue Celebrating the 20th Anniversary of the Alzheimer’s Foundation of America: Understanding and Treating Alzheimer’s Disease.  We confirm that neither the manuscript nor any parts of its content are currently under consideration or published in another journal.

All authors have approved the manuscript and agree with its submission to MDPI.

Response to reviewers second round of comments

            I thank the reviewers for their comments that have improved the previous version of the manuscript.  In the present version it is very clear that this manuscript is NOT and was never intended to be a literature review nor a specific research paper and it is mainly the opinions of single person.   A such I cannot turn this into a literature review as suggested by reviewer 2.  Also, keeping the “telegraphic” or outline style is very important to the structure of the manuscript as a series of challenges to the field.  In the second round of reviews, there were no specific comments except that  “some points need to be enrighed with more bibliography”.  In absence of specific comments, it is hard to implement changes.  The number of references is indeed not uniform as the depth of the literature varies from subject to subject.

            Again, the heart of the issue is whether a manuscript constructed as this is appropriate to Medicina.  Before I submitted it, I did get an unofficial review from one of the editors of this issue that a manuscript of this type might be acceptable.  At this point, I think that the decision is now an editorial one and I am glad to abide by their decision on the current version.

           I thank the reviewers for their very helpful comments.

Sincerely,

Mark Stecker, MD, PhD, FAAN, FACNS, FASNM

Reviewer 3 Report

I am still haviving several concerns about the presentation of results: some points need to be enrighed with more bibiography. After some more revisions it could be published.

Author Response

(The authors gave the same response as above.)
